# Effect of Marine Algae Supplementation on Somatic Cell Count, Prevalence of Udder Pathogens, and Fatty Acid Profile of Dairy Goats’ Milk

**DOI:** 10.3390/ani11041097

**Published:** 2021-04-12

**Authors:** Ferenc Pajor, István Egerszegi, Ágnes Szűcs, Péter Póti, Ákos Bodnár

**Affiliations:** Institute of Animal Husbandry, Hungarian University of Agriculture and Life Sciences, Páter Károly 1, 2100 Gödöllő, Hungary; egerszegi.istvan@uni-mate.hu (I.E.); szucsi.aagi@gmail.com (Á.S.); poti.peter@uni-mate.hu (P.P.); bodnar.akos@uni-mate.hu (Á.B.)

**Keywords:** udder health, mastitis, fatty acids, DHA, *Schizochytrium limacinum*

## Abstract

**Simple Summary:**

Nowadays, there has been increased interest in the modification of the fatty acid composition of foods, such as milk and milk products, to reduce human health problems. The most common way to improve the composition of foodstuffs by n-3 polyunsaturated fatty acids (PUFA) is supplementing animal diets with different plant oils, seeds, fish oil, and freshwater and marine algae. Moreover, fish oil and marine algae (e.g., *Schizochytrium limacinum*) supplements are a good source of long-chain PUFA (LC-PUFA), such as docosahexaenoic acid (DHA). DHA is essential for the development and normal function of the brain, and DHA has beneficial effects for human health, such as reducing the risk of coronary heart disease. In addition, DHA fatty acid has an anti-inflammatory effect, which may help to improve the health of mammary gland secretory activity against the mastitis pathogens. Mastitis pathogen bacterial infection causes an inflammatory reaction within the udder and it leads to reduced milk secretory activity and produces a disadvantageous quality of milk. It was found that marine algae supplementation improved the concentrations of beneficial fatty acids of milk, including higher concentrations of DHA and rumenic fatty acids. In addition, marine algae supplementation decreased the somatic cell counts and prevalence of mastitis pathogens in milk.

**Abstract:**

The aim of this study was to evaluate the effect of the *Schizochytrium limacinum* marine algae on the milk composition and fatty acid profile, somatic cell count, and prevalence of pathogen bacteria in the raw milk of multiparous Alpine goats. Twenty-eight dairy goats were randomly allocated to two groups: control group (C)—fed with 1500 g alfalfa hay and 600 g concentrate; experimental group (MA)—received the same forages and concentrate supplemented with 10 g/head/day marine algae. The goats were housed indoors, while the experiment lasted five weeks, and the milk samples were taken every week. Marine algae feeding had no negative effect on milk composition. The marine algae inclusion significantly decreased the milk somatic cell count and the presence of udder pathogens in the MA group. Mean somatic cell count and presence of udder pathogens were 5.73 log cells/mL and 31%, respectively, in the C group, while these values were 5.34 log cells/mL and 10%, respectively, in the MA group. The marine algae supplementation significantly increased DHA and rumenic acid concentration in the milk of the MA group (0.32 and 0.99 g/100 g of fatty acids, respectively) compared to the C group (0.04 and 0.65 g/100 g of fatty acids, respectively). It can be concluded that a diet supplemented with marine algae significantly improves the udder health of goats and the concentrations of health-promoting fatty acids in milk.

## 1. Introduction

Recently, there has been a significantly increased interest in the modification of the fatty acid composition of animal origin products such as milk and milk products. Enriching the end products by high content n-3 polyunsaturated fatty acids (PUFA) feed supplements (such as oils, seeds, or freshwater and marine algae) is one of the most popular ways for producers [1,2,3]. One of the most interesting fatty acids is docosahexaenoic acid (DHA), which is contained in fish oil and marine algae in large-scale amounts. [4]. DHA has beneficial effects for human health, such as reducing the risk of coronary heart disease [5]. Moreover, DHA is essential for the functional development of the brain in infants and normal brain function in adults; in addition, DHA in the human diet improves learning ability [6]. Presently, the recommended eicosapentaenoic acid (EPA) + docosahexaenoic acid (DHA) daily intake for adults is 250 mg by the European Food Safety Authority (EFSA) Panel on Dietetic Products, Nutrition, and Allergies [7]. The marine algae inclusion in animals’ diet has significantly increased the DHA concentration in milk [8,9]. However, some studies reported that a ruminant diet enriched with marine algae (e.g., *Schizochytrium limacinum*) resulted in decreased dry matter intake, milk yield, and milk fat depression in dairy animals [10,11]. Moreover, authors reported that marine algae feeding has a negative effect on rumen viability, typically decreasing the mass of protozoa and cellulolytic bacteria, and concentrations of odd-chain fatty acids (OCFAs) in milk; these fatty acids are good indicators of ruminal fermentation [4,8,12].

DHA fatty acid has another advantageous effect: it has antioxidant and antihemolytic properties [13]. This fatty acid plays a role in anti-inflammatory processes and in the viscosity of cell membranes [14,15]. The antioxidants delay or inhibit cellular membrane damage mainly through their free radical scavenging property [16,17]. When mastitis occurs, a high amount of free radicals is formed during pathogen phagocytosis. In this way, damaging the mammary epithelial cell, it leads to reduced milk secretory activity in the udder [18]. Mastitis pathogen bacterial infection causes an inflammatory reaction within a dairy animal’s udder. Mastitis is one of the most important diseases of dairy animals. Harmon [19] reported that somatic cell count (SCC) and bacteriological examination indicate the status of the mammary gland as SCC in milk increases during intramammary infection. To the best of our knowledge, there are no studies using *Schizochytrium limacinum* marine algae in diet to improve udder health of goats. We hypothesized that marine algae supplementation in the goat diet increases the bioactive compounds, such as DHA and rumenic acids, and improves the udder health status of dairy goats while serious negative effects on rumen function can be avoided. Previously, daily 15 g/head marine algae supplementation improved the milk fatty acid profile, but it had a slightly negative effect on rumen function. Thus, this study was undertaken to investigate the effect of 10 g/head/day marine algae feeding on short-chain and odd-chain fatty acids as indicators of rumen function, and most favorable bioactive compounds, as well as milk somatic cell counts and prevalence of udder pathogens in goat milk.

## 2. Materials and Methods

### 2.1. Experimental Design

The animal care was in accordance with the guidelines on the protection of animals used for scientific purposes [20]. The study was carried out in an Alpine goat farm (Nógrád County, Hungary; geographical coordinates: 47°85′34.96″ N 19°18′77.50″ E).

From a herd of about 40 polled Alpine goats, 28 homogeneous milking does were selected (days in milk (DIM) 161 ± 3.61), which were balanced for parity (2–3 number of lactation), time of kidding (March), kid rearing (8 weeks), and udder traits (similar udder conformation), and had no signs of clinical mastitis symptoms (swelling, heat, redness, or pain). All animals were kept in full indoor confinement. After weaning, all goats were milked twice a day by machine milking and individually recorded each day. Twenty-eight goats were randomly allocated to two treatment groups (milk yield and body weight in control group: 1.26 kg/d and 59.4 kg, respectively; in experimental group: 1.27 kg/d and 60.6 kg, respectively). The animals in the first group (control group, C, *n* = 14) were fed 1500 g alfalfa hay and 600 g concentrate (ingredients of concentrate, g/kg: maize grain, 300; winter wheat, 150; soybean hull, 150; sugar beet pellets, 100; sunflower meal, 100; soybean meal, 70; sodium bicarbonate, 50; calcium carbonate, 50; salt, 30); in the second group (MA, *n* = 14), goats received the same forages and concentrate with 10 g/head/day dried *Schizochytrium limacinum* marine algae supplementation. The whole investigation period lasted 35 days. The dried microalgae supplement was produced by Alltech Inc. (ALL-G-RICH^®^; Dunboyne, Co Meath, Ireland) (chemical composition: dry matter (DM): 929 g/1000 g, crude protein: 148 g/kg DM, crude fat: 482 g/kg DM, crude fiber: 23 g/kg DM, ash: 38 g/kg DM). In each group, the concentrate was given individually twice a day in morning and evening during milking, while alfalfa hay was offered to the animals twice a day in two equal parts after milking in barns. A commercial trace-mineralized salt block and drinking water were provided free of choice to all animals. The control and the experimental concentrates were approximately isonitrogenous and balanced by energy content. The diets were adjusted to the National Research Council recommendations of energy and protein requirements for dairy goats [21]. The composition of the experimental diets is shown in Table 1.

### 2.2. Collection of Samples

All goats were milked twice a day at 06:00 and 17:00 by a milking machine. Individual milk samples were collected at day 0 (as pretreatment), 7, 14, 21, 28, and 35 into a 50 mL plastic tube for chemical composition and somatic cell counts analysis and into a 10 mL plastic tube for bacteriological examination (only at evening). Moreover, 21 and 35 days from the start of the experiment, milk samples were taken for fatty acid analysis. Previous to milk sampling, the teats of goats were cleaned with antiseptic wipes, and individual milk samples were taken aseptically after the first three milk jets were discarded. Then, all samples were immediately transported to the laboratory. Milk samples were stored at 4 °C for the latter analysis, except samples for fatty acid analysis, which were frozen and stored at −80 °C prior to analysis.

### 2.3. Chemical Analysis

Samples from alfalfa hay, concentrate, and marine algae were taken at the start of the trial and were analyzed for dry matter, crude protein, crude fat, crude fiber, and crude ash according to the procedure of the Hungarian Feed Codex [22].

Milk samples (*n* = 168) were analyzed for identification of udder pathogens bacterium species. Milk (0.1 mL) was plated on Columbia esculin blood agar (Biolab Inc, Budapest, Hungary) containing 5% sheep blood and 0.5% esculin, and incubated at 37 °C for 48 h. The isolates were identified as pathogen udder species by conventional methods, including Gram staining, colony morphology, and hemolysis patterns according to the National Mastitis Council guidelines [23]. The morning and evening samples were mixed for each goat before analysis. The milk somatic cell count was determined using the LactoScan SCC apparatus (Milkotronic Ltd., Nova Zagora, Bulgaria).

Fat, protein, lactose, and total solids contents of milk were determined using the LactoScope™ IR spectrometry analyzer (Delta Instruments, Drachten, The Netherlands).

The alfalfa, concentrate, marine algae meal, and milk fat were extracted with the method of Gerber [24]. Fatty acids (FA) were re-esterified to methyl esters using the procedures according to ISO 12966-2 (2011) standard [25]. Methyl esters of FA were determined by gas chromatography (gas chromatographer GC 2010, Shimadzu, Kyoto, Japan) with a flame ionization detector (FID) and a column (Zebron ZB-WAX, 30 m × 0.25 mm × 0.25 μm). The split injection ratio was 50:1. Helium was used as the carrier gas, applying a flow rate of 28 cm/s. The injector and detector temperatures were 270 and 300 °C, respectively. The oven temperature programmed run started at 80 °C, then was increased 2.5 °C/min up to 205 °C and held for 20 min, and then increased again to 225 °C at 10 °C/min and held for 5 min. Peaks were identified on the basis of the retention times of standard methyl esters of individual FA (Supelco 37 Component FAME Mix, Sigma-Aldrich, St. Louis, MO, USA). The individual FA were calculated by the ratio of their peak area to the total area of all observed acids. The selected FA combinations were calculated by using FA data: saturated fatty acids (SFA); monounsaturated fatty acids (MUFA); polyunsaturated fatty acids (PUFA); total n-6 and n-3 PUFA and n-6/n-3 ratio. Atherogenic index (AI) was calculated according to Ulbricht and Southgate [26]. DHA transfer from feed to milk efficiency was calculated according to Moate et al. [1]: DHA in milk yield (mg/day)/DHA intake (mg/day).

### 2.4. Statistical Analysis

Statistical analysis using the general linear model (GLM) method was processed by the SPSS 25.0 software package. Shapiro–Wilk’s test was used for testing the normality distribution. The somatic cell counts data were transferred to logarithm (with logarithm base 10) before statistical procedures. The prevalence of udder pathogens between treatments (control and marine algae groups) was determined using the Chi2 test. The general linear model (GLM) for repeated measures analysis of variance was explored. The effect of diet (C and MA), sampling time (7, 14, 21, 28, and 35 days for milk composition and somatic cell counts and 21 and 35 days for FA composition), and the interaction of diet and sampling time on milk composition, somatic cell counts, and fatty acid composition were determined. The statistical model was as follows:y_ijk_ = μ + D_i_ + S_j_ + (D × S)_k_ + e_ijk_(1)
where y_ijk_ is the value of the dependent variable, μ is the overall mean, D_i_ is the effect of diet, S_j_ is the effect of sampling time, (D × S)_k_ is the effect of the interaction of diet and sampling time, and _eijk_ is the random error. The Least Significant Difference (LSD) post hoc test was used for pairwise comparisons. Differences are shown when *p* < 0.05.

## 3. Results

The marine algae supplementation did not affect the chemical composition of milk (Table 2).

Under the experiment, the milk composition was consistent between the two groups. The mean value of the milk yield was 1.25 kg for the experimental group (MA) and 1.30 kg for the control group (C). In contrast, marine algae inclusion in the animals’ diet had a significant effect on the milk somatic cell counts. At pretreatment, milk somatic cell counts were not different between C (5.63 log cell/mL) and MA (5.58 log cell/mL) groups. However, marine algae feeding significantly decreased the milk somatic cell counts in milk samples in the MA group, while the milk somatic cell counts were consistent in the C group. The mean value of the somatic cell counts during experimental treatment was 5.34 log cell/mL, while the mean value of somatic cell counts was 5.73 log cell/mL in the C group (*p* < 0.001). Marine algae feeding affected the somatic cell counts of milk (Figure 1). At the first week of the experimental period, the somatic cell counts were not changed (5.61 log cell/mL), then from the second week, the mean somatic cell counts were dramatically decreased to about 200 thousand cells/mL (5.17–5.31 log cell/mL). During the remaining period of the treatment, the milk somatic cell counts were consistent.

Marine algae supplementation had a significant effect on the prevalence of udder pathogens in milk samples (Table 3).

Throughout the experimental period, only the coagulase-negative *Staphylococcus* (CNS) mastitis pathogen was identified; other types of pathogens in the milk samples were not found. At pretreatment, the prevalence of udder pathogens was not different between C (36%) and MA (29%) groups. During treatment, the prevalence of udder pathogens was consistent in the C group (Figure 2).

On the contrary, the enrichment of experimental diets with marine algae significantly decreased the prevalence of udder pathogens in milk samples. In the first week of marine algae feeding treatment, the prevalence of udder pathogens substantially decreased down to 14.3%, and then further decreased down to 7.1% in the second week. In the third week, the prevalence of pathogen bacteria was slightly increased, then decreased down to 7.1% in the fourth week, and this value remained consistent until the end of the experimental period. The mean value of the prevalence of udder pathogens during experimental treatment was about 10%.

The marine algae supplementation had a great impact on milk fatty acid profile (Table 4).

The concentrations of capric acid (C10:0), myristic acid (C14:0), myristoleic acid (C14:1), palmitic acid (C16:0), palmitoleic acid (C16:1), vaccenic acid (t11 C18:1), rumenic acid (c9t11 C18:2), arachidonic acid (C20:4), docosahexaenoic acid (C22:6), and saturated fatty acids were significantly increased, while the concentrations of stearic acid (18:0), oleic acid (c11 C18:1), linoleic acid (C18:2), alpha-linolenic acid (C18:3), eicosapentaenoic acid (C20:5), docosapentaenoic acid (C22:5), C16:0/C18:1 ratio, monounsaturated fatty acids, n-6 fatty acids, n-6/n-3 ratio, and atherogenic index in the goat milk were noticeably decreased in the milk fat of marine algae-supplemented goats.

The sampling days had significantly influenced the capric acid (C10:0), lauric acid (C12:0), palmitoleic acid (C16:1), stearic acid (C18:0), oleic acid (c11 C18:1), linoleic acid (C18:2), alpha-linolenic acid (C18:3), eicosapentaenoic acid (C20:5), docosahexaenoic acid (C22:6), C16:0/C18:1 ratio, saturated fatty acids, and monounsaturated fatty acids in milk.

The interaction between diet and sampling days had significantly altered the capric acid (C10:0), myristic acid (C14:0), myristoleic acid (C14:1), palmitic acid (C16:0), palmitoleic acid (C16:1), rumenic acid (c9t11 C18:2), alpha-linolenic acid (C18:3), arachidonic acid (C20:4), eicosapentaenoic acid (C20:5), docosahexaenoic acid (C22:6), C16:0/C18:1 ratio, saturated fatty acids, and atherogenic index in milk.

The daily DHA intake was 1352.29 mg throughout the experimental period. Daily 10 g/head marine algae supplementation resulted in 9.06 and 13.32 mg DHA content in 100 mg of milk in the MA group at the 21st and 35th days of experiment, respectively (Table 5). The DHA conversion efficiency ratio from marine algae to milk was 7.97% and 11.73% at the 21st and 35th days of the treatment, respectively.

## 4. Discussion

Values of milk composition parameters were within the normal ranges for dairy goats reported by several authors [27,28]. The fat and protein content of the milk was not influenced by marine algae feeding. Lack of significance of milk fat and protein content may be due to the low amount of supplemented marine algae (10 g/head/day). This corresponds with previous reports, where marine algae diet contains daily 105 g/head/day for dairy cows [29], and 15g/head/day for dairy goats [8].

In contrast, Boeckaert et al. [30] reported that an algae supplementation level of about 10 g/kg DM intake significantly reduced the cow milk fat content. Moreover, Bichi et al. [11] and Mavrommatis and Tsiplakou [9] found milk fat depression in dairy ewes and dairy goats fed algae-containing fodder (8 g/kg DM and from 10 to 36 g/kg DM, respectively). Moreover, other authors reported that the milk fat and protein content were improved by feeding marine algae. Papadopoulos et al. [31] and Reynolds et al. [32] reported that ewe milk fat and protein contents were increased by a marine algae-containing diet (main doses: 23.5, 47, and 94 g/day/head; and 10 g/kg DM/day/head). Milk fat and mainly protein content are important for farmers and milk processors, because it is well known that these components have a great effect on cheese composition and yield.

The somatic cell counts were lower in animals fed the marine algae-containing diet in comparison to the group fed the control diet. These results may indicate an increase in anti-inflammatory processes that resulted in enhanced mammary gland health for goats fed marine algae. The earlier reported recommended limit of somatic cell count of healthy goats’ milk is 1 million cells/mL [27,33]. Below this limit, goats did not show the symptoms of mastitis. The threshold limit for goat milk by the US Food and Drug Administration is also 1 × 10^6^ SCC/mL [34], although, in the EU, there is no available standard limit for somatic cell count in goat milk. In addition, Sramek et al. [35] reported elevated somatic cell counts associated with a reduced secretory activity of the mammary epithelium; for this reason, the unfavorable udder health caused a decreased concentration of de novo fatty acids (from C4:0 to C14:0 and half part of C16:0) in milk fat.

The mean value of prevalence of udder pathogens in both groups shows favorable values, in correspondence with what was earlier reported by Bagnicka et al. [36], but markedly lower than results of [37,38,39]. However, during treatment, the prevalence of udder pathogens was significantly different in the two groups. Marine algae feeding dramatically decreased the prevalence of udder pathogens in milk samples. Under treatment, the mean values of udder pathogens in the MA and C groups were 10% and 31%, respectively. All minor mastitis pathogens were CNS; this was similar to the results of Bagnicka et al. [36] and Souza et al. [40], who found that the most frequent minor udder pathogen was CNS. These authors stated that prevalence of CNS mastitis pathogens caused the increase of the total somatic cell count in dairy goats. In our study, major mastitis pathogens in milk samples were not found. This type of udder pathogen bacteria has great impact on udder health; as Contreras et al. [41] reported, major types of udder pathogens caused clinical mastitis in dairy animals. When udder pathogens infection occurs, due to high amount of free radicals, which are formed during pathogen phagocytosis, the mammary epithelial cells are damaged, and as a result, milk secretory activity in the udder is decreased [18]. Our results suggested that the DHA fatty acid incorporated into phospholipids of the mammary gland membrane and improved the mammary epithelial cells, due to reduced inflammatory reaction in the udder. It is well known that DHA fatty acid plays a role in anti-inflammatory processes and improves cell membranes [14,15]. Earlier reports found that high n-3 PUFA intake has a beneficial effect on udder health. Košmelj et al. [42] and Gantner and Kompan [43] found that α-linoleic fatty acid-enriched diet of goats had a significant effect on lower somatic cell count in milk.

The fatty acid concentrations changed markedly within three weeks and remained relatively constant under the experimental period. These results were in concordance with the results of earlier reports of [1,44,45].

The marine algae supplementation did not affect the concentrations of short- and medium-chain saturated fatty acids (SMCSFA) (from C4:0 to C12:0), except for capric acid (C10:0). The concentration of capric acid was significantly higher in the MA group compared to the C group. An earlier study [46] reported that the capric acid has a significant effect on the flavor of dairy products. The organoleptic properties of milk are important for consumers, however, during this study, these traits were not evaluated. In further research, the effect of MA supplementation on milk organoleptic properties should be tested. In addition, the medium-chain saturated fatty acids are becoming more and more interesting for nutritionists. The consumption of these fatty acids causes slight weight loss without any negative effect on lipid metabolism [47]. Gómez-Cortés et al. [48] summarized the SMCSFA’s potential benefits in human health in a review report.

The marine algae-enriched diet elevated the concentrations of myristic acid (C14:0) and palmitic acid (C16:0), due to the high concentration of these fatty acids in the experimental diet. This is in concordance with the results of Mavrommatis and Tsiplakou [9].

In addition, the odd-chain fatty acids (OCFAs) concentrations remained constant in the milk of marine algae-fed goats. OCFAs mostly originated from the rumen bacterial populations [49]. These fatty acids are a good indicator of the health of ruminal bacterial populations. The odd-chain fatty acids absorbed by the intestinal wall and taken up by the mammary gland from blood plasma led to the presence of odd-chain FA in milk fat, similar to long-chain fatty acids. However, other studies reported that the marine long-chain PUFA supplements in diets of ruminants influenced negatively the microbiota composition (e.g., cellulolytic bacteria) in rumen, causing milk fat depression [4,12] and linking with the lower odd-chain fatty acids (OCFAs) concentrations in milk. To sum up, the results show that the daily 10 g/head marine algae supplementation had no negative effect on rumen viability and rumen fermentation.

The contents of stearic acid and oleic acid were significantly lower in the experimental group. Daily 10 g/head marine algae inclusion in the goats’ diet had a great impact on biohydrogenation in the rumen. The long-chain PUFA n-3 in the diet inhibits the polyunsaturated fatty acids saturation to C18:0 and various isomers of C18:1 in the rumen [10]. Therefore, the feeding of marine algae decreased markedly the stearic acid in milk fat. The reducing availability of stearic acid greatly decreased the oleic acid (c11 C18:1) concentration in milk. Shingfield et al. [50] reported that the stearoyl-CoA desaturase enzyme was responsible for cc. 60% of the amount of oleic acid synthesis in milk, while the other part originated from the digestive tract.

In addition, the ratio of oleic acid to palmitic acid sharply decreased in the MA group. The ratio of oleic acid to palmitic acid was 0.82 in the C group, while this value was 0.57 in the MA group (*p* < 0.01). This ratio is important for milk producers, due to the higher ratio of oleic acid to palmitic acid resulting in reduced starter culture activity in cheese [51].

In the milk of marine algae-fed goats, a significant increase in rumenic acid (by 52%) was found. An earlier study reported that the long-chain n-3 PUFAs in the diet inhibit vaccenic acid saturation in the rumen [10]. In this way, the feeding of marine algae increased markedly the vaccenic fatty acid content in milk. The vaccenic acid is the primary precursor of rumenic acid; vaccenic acid is converted to rumenic acid by Δ9-desaturase in the mammary gland [52]. Rumenic acid has been shown to suppress carcinogenesis, modify the immune system, and reduce atherogenesis [53,54].

Concentration of DHA, which is required for many metabolic processes and has many positive effects on human health, was increased by the marine algae-enriched diet. In the present study, the mean DHA value in the MA group increased fatty acids up to 0.38 g/100 g from 0.04 g/100 of fatty acids; the difference is more than ninefold. Toral et al. [45] and Pajor et al. [8] also found that feeding marine algae supplements considerably increased the DHA content in milk. In the present study, the mean values of DHA transfer efficiency in marine algae treatments at 21 and 35 days were 7.97% and 11.73%, respectively. Previously, Moran et al. [28] found a similar tendency; the DHA transfer efficiency from algae to milk markedly raised up to 5.71% under 21 days, then slightly reached the top value (around 7.71%) two weeks later. In this work, the mean value of the DHA transfer efficiency under the experimental period was 6.96%. Other studies reported that DHA transfer efficiency was between 8% and 12% in different treatments [1,8,44,55]. However, the DHA transfer efficiency ratio from a fish oil-supplemented diet to milk was less than 5% in cows [56].

In addition, a significant decrease in the n-6/n-3 ratios was observed in the experimental goats receiving marine algae. This is consistent with earlier reports [8,9]. The n-6/n-3 ratio is generally used to assess the nutritional value of fats. The lower n-6/n-3 ratio in the milk of goats fed *Schizochytrium limacinum* marine algae is related to the new recommendations for human nutrition [7]. Moreover, the atherogenic index (AI) was improved by marine algae supplementation in the experimental treatment. The MA diet significantly decreased the AI in milk (2.95 vs. 2.45; *p* < 0.001). The lower AI value in the milk of the MA group met the new recommendations for human nutrition.

## 5. Conclusions

Based on the results of this study, it may be concluded that feeding 10 g/head/day marine algae supplementation had a strong effect to reduce the presence of udder pathogens and somatic cell count during the first five weeks of the feeding period. Daily 10 g/head *Schizochytrium limacinum* marine algae intake in the goats’ diet did not affect their milk yield, milk composition, and odd-chain fatty acids concentration in milk. In contrast, marine algae supplementation significantly increased the rumenic acid and docosahexaenoic fatty acid concentrations in milk. Supplementation with marine algae is not only suitable for improving the content of bioactive compounds in milk, but improves the udder health of goats.

## Figures and Tables

**Figure 1 animals-11-01097-f001:**
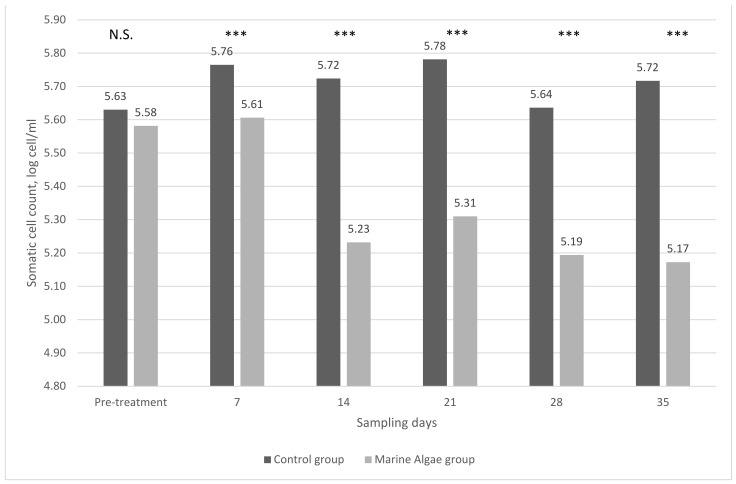
Somatic cell counts of goat milk from different feeding treatments (each group and sampling day *n* = 14). N.S.—not significant; ***— *p* < 0.001.

**Figure 2 animals-11-01097-f002:**
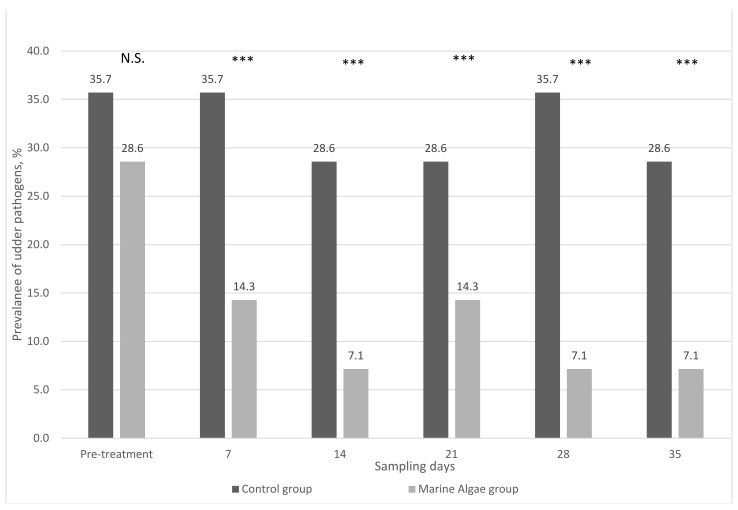
Prevalence of udder pathogens from different feeding treatments (each group and sampling day *n* = 14). N.S.—not significant; ***— *p* < 0.001.

**Table 1 animals-11-01097-t001:** Chemical composition and fatty acid (FA) profile of fed forage.

Items	Diet	
	Control ^1^	Marine Algae ^2^
Daily intake, g		
alfalfa hay	1500	1500
concentrate	600	600
marine algae ^3^	− ^4^	10
Ingredients, DM% ^5^		
alfalfa hay	71.88	71.53
concentrate	28.12	27.98
marine algae ^3^	−	0.49
DM intake, kg/day	1.88	1.89
Chemical composition		
dry matter, g/kg forage	894.29	894.45
crude protein, g/kg DM	198.57	198.32
crude fat, g/kg DM	22.82	25.08
crude fiber, g/kg DM	217.25	216.29
crude ash, g/kg DM	77.77	77.57
NEl ^6^, MJ/kg DM	6.17	6.18
Main FA, g/100g of fatty acids		
C12:0	0.21	0.24
C14:0	0.58	1.21
C16:0	13.17	17.52
C18:0	2.73	2.67
C18:1n-9	30.33	27.53
C18:2n-6	32.04	29.02
C18:3n-3	15.62	14.15
C22:6n-3 (DHA) ^7^	−	2.85

^1^ Control—control diet (hay and concentrate); ^2^ Marine algae—control diet supplemented with 10 g/head/day microalgae; ^3^ Marine algae contained g/100 g of fatty acids: C12:0 0.5, C14:0 7.28, C16:0 59.10, C18:0 2.04, C18:1n-9 0.71, C18:2n-6 0.13, C18:3n-3 0.10, C22:6n-3 30.10; ^4^ −: not contained; ^5^ DM—dry matter; ^6^ NEl—net energy for lactation; DHA: docosahexaenoic acid; ^7^ DHA daily intake: 1352.29 mg.

**Table 2 animals-11-01097-t002:** Milk yield and chemical composition and somatic cell counts of goat milk from different feeding treatments all through the experimental period

Traits	Pretreatment		Diet		Sampling					SEM	*p*-Value		
	C	MA	C	MA	7	14	21	28	35		D	S	D × S
Milk yield, kg	1.26	1.27	1.25	1.30	1.28	1.25	1.26	1.30	1.28	0.017	0.171	0.897	0.948
Fat, %	3.92	3.83	3.71	3.74	3.75	3.63	3.62	3.87	3.74	0.035	0.675	0.159	0.991
Protein, %	3.41	3.49	3.40	3.44	3.44	3.34	3.50	3.51	3.32	0.031	0.493	0.191	0.889
Lactose, %	4.41	4.45	4.45	4.44	4.41	4.45	4.44	4.45	4.46	0.011	0.753	0.732	0.527
Total solids, %	12.43	12.49	12.32	12.44	12.35	12.34	12.34	12.50	12.38	0.052	0.245	0.834	0.879
SCC, log cell/mL	5.63	5.58	5.73	5.34	5.69 ^a^	5.54 ^b^	5.61 ^b^	5.47 ^b^	5.52 ^b^	0.035	0.000	0.000	0.608

C—control diet (hay and concentrate), MA—control diet supplemented with 10 g/head/day marine algae, SCC—log somatic cell counts, ^a, b^—means with different letters differ significantly at *p* < 0.05.

**Table 3 animals-11-01097-t003:** Prevalence of udder pathogens in milk samples from different feeding treatments (%).

Mastitis Pathogens	Pretreatment		*p*-Value	Treatment		*p*-Value
	**C (*n* = 14)**	**MA** **(*n* = 14)**		**C (*n* = 70)**	**MA** **(*n* = 70)**	
Negative	64 (*n* = 9)	71 (*n* = 10)	0.365	69 (*n* = 78)	90 (*n* = 60)	<0.001
Infected samples *	36 (*n* = 5)	29 (*n* = 4)	0.365	31 (*n* = 22)	10 (*n* = 10)	<0.001

C—control group; MA—marine algae group; * Coagulase-negative *Staphylococcus* (CNS) pathogens were identified from all infected samples.

**Table 4 animals-11-01097-t004:** Fatty acid profile of goat milk from different feeding treatments (g/100 g of fatty acids).

Fatty Acids	Diet		Sampling Days		SEM	*p*-Value		
	C	MA	21	35		D	S	D × S
C4:0	1.46	1.44	1.45	1.45	0.009	0.176	0.984	0.320
C6:0	1.19	1.17	1.17	1.19	0.019	0.624	0.637	0.777
C8:0	1.83	1.84	1.82	1.86	0.034	0.971	0.590	0.320
C10:0	7.20	8.29	7.77	8.23	0.076	0.000	0.004	0.030
C12:0	4.82	4.70	4.52	5.00	0.116	0.617	0.043	0.179
C14:0	10.87	11.88	11.36	11.40	0.124	0.000	0.880	0.004
C14:1	0.19	0.25	0.22	0.23	0.006	0.000	0.284	0.001
C16:0	29.44	34.66	32.29	31.82	0.385	0.000	0.546	0.011
C16:1	0.56	0.68	0.59	0.65	0.003	0.000	0.000	0.000
C18:0	9.54	6.59	7.34	8.79	0.193	0.000	0.000	0.075
c11 C18:1n-9	23.63	18.64	20.13	22.13	0.217	0.000	0.000	0.653
t11 C18:1n-7	1.39	1.68	1.49	1.58	0.021	0.000	0.054	0.298
rumenic acid	0.65	0.99	0.81	0.83	0.014	0.000	0.454	0.002
C18:2n-6	3.33	3.06	3.21	3.18	0.054	0.016	0.731	0.697
C18:3n-3	1.18	0.95	1.11	1.02	0.014	0.000	0.003	0.001
C20:3n-6	0.03	0.03	0.03	0.02	0.001	0.328	0.001	0.328
C20:4n-6	0.18	0.20	0.19	0.19	0.003	0.005	0.517	0.009
C20:5n-3 (EPA)	0.09	0.07	0.09	0.08	0.002	0.000	0.001	0.007
C22:5n-3	0.22	0.18	0.20	0.20	0.004	0.000	0.707	0.065
C22:6n-3 (DHA)	0.04	0.32	0.16	0.21	0.004	0.000	0.000	0.000
palmitic/oleic ratio	0.82	0.54	0.64	0.72	0.012	0.000	0.001	0.044
odd FA	2.28	2.26	2.26	2.27	0.020	0.782	0.950	0.224
SFA	67.46	71.08	68.25	70.28	0.438	0.000	0.024	0.003
MUFA	25.81	21.29	22.47	24.63	0.220	0.000	0.000	0.482
PUFA	5.77	5.82	5.84	5.76	0.061	0.658	0.533	0.233
n-6	3.56	3.30	3.45	3.41	0.055	0.021	0.730	0.582
n-3	1.56	1.54	1.58	1.52	0.018	0.632	0.107	0.837
n-6/n-3 ratio	2.32	2.14	2.20	2.26	0.039	0.028	0.440	0.413
AI	2.95	2.48	2.63	2.79	0.041	0.000	0.057	0.000

C—control group; MA—marine algae group; AI: atherogenic index calculated by Ulbricht and Southgate [25].

**Table 5 animals-11-01097-t005:** Calculated docosahexaenoic acid (DHA) conversion efficiency from diet to milk.

Days ^1^	Daily Intake of DHA, mg/Day	Average Milk DHA Content, mg/100 g Milk	Milk Production, kg	DHA in Milk Yield, mg/Day	DHA Efficiency Ratio ^2^, %
21	1352.29	9.06	1.19	107.80	7.97
35	1352.29	13.32	1.19	158.62	11.73

^1^ control diet supplemented with 10 g/head/day marine algae; ^2^ calculated according to Moate et al. [1].

## Data Availability

Data are available from the corresponding author upon request.

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
