# Peer review of "Effect of Marine Algae Supplementation on Somatic Cell Count, Prevalence of Udder Pathogens, and Fatty Acid Profile of Dairy Goats’ Milk"

_animals, 2021, doi:10.3390/ani11041097_

Round 1
Reviewer 1 Report
The authors presented an interesting work on the supplementation of goats. Marine algae are a rich source of long chain fatty acids. The authors demonstrated a beneficial effect of supplementation on milk composition and animal health. However, while reading, some doubts arise.
Simple Summary should not be an introduction to Abstract. Reading the Simple summary proposed by the authors, I got the impression that I received a review paper for review.
L 33-34 - "… .improved the hygiene…." - does it help in cleaning and disinfection processes? Or maybe it refers to the estimated frequency of Staphylococcus spp. Occurrence? The term is imprecise.
Results:
Table 2 is unreadable - first, you can only guess that the values of milk composision are expressed in% and SCC in log cell / ml. Second, the values given in the "Diet" column are mean values or maybe determined after the end of the experiment? We can only guess.
The authors declare that they will assess the prevalence of udder pathogens, and they only mean Staphylococcus spp bacteria. These are not the only pathogens found in milk and not the only ones causing mastitis. I propose to clarify this and edit the text.
L 199-207 - the values given in the text are not exact - in my opinion they should be given exactly or labeled "about"
L 262 - the term "lack of major mastitis pathogens" is not valid in the context of the presented work. According to the authors of "major mastitis pathogens (such as S. aureus) in milk samples were not found", we do not know whether the lack of these pathogens was caused by the proposed supplementation or the high health standard of the herd in which the experiment was carried out.
The authors observed a significantly higher level of capric acid in the milk of supplemented goats. Whether the organoleptic tests of the obtained milk were carried out, it would be interesting if the supplementation proposed by the authors changes the organoleptic properties of milk, which is a very important factor in the raw material for the production of dairy products.
Author Response
"Please see the attachment."

Reviewer 2 Report
The aim of Pajor et al. study is very interesting because of the importance of reducing udder pathologies and improving milk quality since the point of view of human health. However, I think that paper must be improved slightly. Here are some comments:
Abstract:
L28 and L31: In my opinion, significantly is more accurate than considerably.
Keywords: Please, don’t use words that already appear in title as keywords.
Introduction:
L43-44: Please, modify the construction of this sentence (Contained of this long chain fatty acid is the fish oil and marine algae in large scale amounts [4]) to make it clearer.
L67: Delete first comma.
L70: Why the authors decide to use that quantity of algae? Did they base their decision in previous research? Please, state that in Introduction.
Materials and methods:
L82: Why not select goats by milk yield and body weight too?
L83: Were treatment groups homogeneous?
L85: Please, indicate the ingredients of concentrate.
L159: Please, define I, j and k.
Results:
L182: 200 thousands? Please, be more accurate.
Figures: I recommend to use colours.
Discussion:
L215. Miristoleic acid is C14:1
L216. Please, explain which fatty acids are de novo fatty acids.
L222: Please delete had.
Table 4. I think that it would be interesting to add other indices related to milk quality since the point of view of human health, as Atherogenic Index or Thrombogenicity Index, to emphasize the importance of n3 enriched milk against the coronary diseases.
Table 5. There are any comments of recording feed intake in Material and Methods. So, please, include an explanation of that and a statement about feed consumption in Results.
Author Response
"Please see the attachment."

Reviewer 3 Report
I have the following comments and suggestions for editing the manuscript:
1) I recommend deleting "Schizochytrium limacinum" in the title of the article. Name edit to: Effect of marine algae supplementation on somatic cell count, prevalence of udder pathogens and fatty acid profile of dairy goats' milk. The title of the paper is, in my opinion, too long and it is sufficient if a specific type of algae is listed in the methodological section.
2) On line 15 and lines 44 and 45, I recommend adding the text, as DHA also has other, perhaps more significant, effects than just on coronary heart disease. It is generally known that: docosahexaenoic acid (DHA) is essential for the growth and functional development of the brain in infants. DHA is also required for maintenance of normal brain function in adults. The inclusion of plentiful DHA in the diet improves learning ability, where deficiencies of DHA are associated with deficits in learning.
3) In the Abstract, line 34, I recommend changing the term "hygiene" (inaccurate) to "udder health and quality of milk".
4) Line 161 shows the value for DIM 161. Does this mean that all 28 goats were kidded on the same day? If not, in addition to the average value (161), the standard deviation must be stated.
5) I recommend adding in Table 2 the data on the daily milk production of goats during the experiment (add the appropriate line before the chemical composition of the milk), given that the Conclusions chapter (line 358) states that “... marine algae intake on the goats 'diet did not affect their milk yield”. Based on the proposed adjustment, the text on lines 165 and 166 also needs to be corrected. In Table 2, the units (g / 100g or %) also need to be added. I recommend using the term Log10SCC instead of "SCC" in Table 2, or explaining under Table 2 that "SCC" is log10SCC or log SCC.
6) On line 179, I recommend adding after the text: "... in C group (P <0.001)".
7) Table 3 and also in Figure 2 show the number of negative and infectious samples in %. It would be appropriate to indicate in the methodological part of the work how many samples in absolute numbers were analysed in the "pre-treatment" phase (day 0) and during the experiment itself (days 7 to 35). The number of analysed samples affects the reliability of the obtained results.
8) In Table 4 it is necessary to add in which units the values of individual fatty acids are given and to harmonize it with the text (e.g. on lines 32, 338 and 339 the values are given in %, on line 101 in g / 100 g FA). In scientific papers with a similar focus, fatty acid values are given differently, for example as mg / 100 mg FAME, or g. 100 g-1 FAME (or fatty acids).
9) Lines 216 and 229 state the effect on "de novo fatty acids". Such a group of fatty acids is not listed in Table 4. Which fatty acids fall into this group is listed only on line 267. I recommend adding to the methodological part of the manuscript or modifying the chapter Results.
10) On line 299, "... on weight control ..." is incorrect. The cited work speaks of "weight loss". The sentence needs to be modified.
11) There is no clear sentence on lines 345 and 346 (..under 7-84 days ???).
Proposed minor edits to the text:
- Grammatical correction of the sentence on lines 43 and 44 is required.
- In equation (1) it should be: yij = µ + Di + Sj + (DxS)k + eijk. The text on lines 159 and 160 should also be corrected accordingly.
- On lines 161 and 162, the sentence "In the case ..." must be deleted - the text is repeated - see line 153.
- On lines 248 and 249, I recommend styling the sentence.
- In line 303, edit the sentence: "... (2020)".
- In line 306, edit the sentence "... Harfoot, 1981)".
Author Response
"Please see the attachment."

Round 2
Reviewer 1 Report
Thank you for the answer and explanations regarding the presented work.
In its current form, I believe the work can be published.